# Distinct Pattern of Endoplasmic Reticulum Protein Processing and Extracellular Matrix Proteins in Functioning and Silent Corticotroph Pituitary Adenomas

**DOI:** 10.3390/cancers12102980

**Published:** 2020-10-14

**Authors:** Alexander K. Eieland, Kjersti R. Normann, Arvind Y. M. Sundaram, Tuula A. Nyman, Kristin A. B. Øystese, Tove Lekva, Jens P. Berg, Jens Bollerslev, Nicoleta C. Olarescu

**Affiliations:** 1Section of Specialized Endocrinology, Department of Endocrinology, Oslo University Hospital (OUS), 0424 Oslo, Norway; aleeie@ous-hf.no (A.K.E.); k.r.normann@medisin.uio.no (K.R.N.); k.a.oystese@medisin.uio.no (K.A.B.Ø.); jens.bollerslev@medisin.uio.no (J.B.); 2Research Institute for Internal Medicine, Oslo University Hospital (OUS), 0372 Oslo, Norway; Tove.Lekva@rr-research.no; 3Institute of Clinical Medicine, Faculty of Medicine, University of Oslo, 0318 Oslo, Norway; t.a.nyman@medisin.uio.no (T.A.N.); j.p.berg@medisin.uio.no (J.P.B.); 4Department of Medical Genetics, University of Oslo, Oslo University Hospital (OUS), 0450 Oslo, Norway; arvind.sundaram@medisin.uio.no; 5Department of Immunology, Oslo University Hospital (OUS), 0424 Oslo, Norway

**Keywords:** RNA sequencing, proteomics, cell adhesion, endoplasmic reticulum protein folding

## Abstract

**Simple Summary:**

Corticotroph pituitary adenomas present a spectrum of functionality regarding hormonal production, ranging from functioning to silent tumors. Moreover, they show different invasiveness and recurrent behavior profiles, the silent being considered an aggressive type of adenomas. Through analyses of global transcriptome and proteome, we show that both groups expressed genes and protein related to protein synthesis and vesicular transport, and present a distinct pattern of collagen/ extracellular matrix proteins. Endoplasmic reticulum protein processing is a key factor for hormone production in functioning corticotroph adenomas. Furthermore, a distinct cell adhesion profile in silent corticotroph adenomas may explain the aggressive behavior. Together, our findings shed light on the different repertoires of activated signaling pathways in corticotroph pituitary adenomas and may reveal new potential medical targets.

**Abstract:**

Functioning (FCA) and silent corticotroph (SCA) pituitary adenomas act differently from a clinical perspective, despite both subtypes showing positive TBX19 (TPIT) and/or adrenocorticotropic hormone (ACTH) staining by immunohistochemistry. They are challenging to treat, the former due to functional ACTH production and consequently hypercortisolemia, and the latter due to invasive and recurrent behavior. Moreover, the molecular mechanisms behind their distinct behavior are not clear. We investigated global transcriptome and proteome changes in order to identify signaling pathways that can explain FCA and SCA differences (e.g., hormone production vs. aggressive growth). In the transcriptomic study, cluster analyses of differentially expressed genes revealed two distinct groups in accordance with clinical and histological classification. However, in the proteomic study, a greater degree of heterogeneity within the SCA group was found. Genes and proteins related to protein synthesis and vesicular transport were expressed by both adenoma groups, although different types and a distinct pattern of collagen/extracellular matrix proteins were presented by each group. Moreover, several genes related to endoplasmic reticulum protein processing were overexpressed in the FCA group. Together, our findings shed light on the different repertoires of activated signaling pathways in corticotroph adenomas, namely, the increased protein processing capacity of FCA and a specific pattern of adhesion molecules that may play a role in the aggressiveness of SCA.

## 1. Introduction

Non-functioning pituitary adenomas (NFPAs) comprise up to a half of all pituitary neuroendocrine tumors (PitNETs) [1,2], and although they are characterized as silent, patients present increased morbidity and decreased quality of life due to clinical manifestations caused by the tumor pressure [3,4]. The latest World Health Organization (WHO) classification using immunohistochemistry (IHC) for transcription factors and anterior pituitary hormones illustrates that the most common NFPA subtype is the silent gonadotroph followed by the silent corticotroph adenoma (SCA) [5]. SCA comprise 5–19% of the NFPAs, depending on the cohort, and approximately 3% of all adenomas, whereas their functional counterpart, functioning corticotroph pituitary adenoma (FCA), represent only 2–6% [1,2,6,7]. SCA is categorized as a potentially more aggressive neuroendocrine tumor posing a high risk of increased invasiveness and parasellar growth, a tendency for multiple and earlier recurrences, and higher rates of hemorrhage and apoplexy [8,9,10,11,12]. Conversely, its clinically active counterpart rarely poses challenges due to tumor volume. Due to the presence of clinical symptoms, FCA is discovered at an earlier stage in tumor development, most often as a microadenoma located in the sellar area (in 80% of cases) [13], whereas SCA present as a macroadenoma with extrasellar expansion at the time of diagnosis [14]. Efforts to explain the aggressiveness and the propensity for recurrence of SCA as compared to FCA are, to date, limited and inconclusive [15].

There is a spectrum of functionality of the corticotroph adenoma ranging from totally silent to whispering, and finally to clinically functioning corticotroph adenoma [5]. The molecular mechanism behind the silence of the SCA is yet to be unveiled, though several hypotheses have been proposed.

For example, it has been proposed that SCA presents a dysfunctional processing of ACTH from its precursor protein pro-opiomelanocortin (POMC), whereby a variety of ACTH precursors are secreted [10,13,14,16,17]. POMC is cleaved by different enzymes into a variety of peptides in a tissue-dependent manner [18]. Among these, prohormone convertases, PCSK1 and PCSK2, were shown to have defective expression in SCA, perhaps contributing to the diminished ACTH production [14,19].

Another hypothesis postulates that SCA may contain incompletely differentiated precursors of corticotroph cells, not yet capable of secreting bioactive hormones. This has been reinforced by their lower expression of both the corticotroph transcription factor, TBX19, POMC, and some adenomas showing poor and immature differentiation [13]. Less differentiated corticotroph cells as described in SCA might also imply that their secretory machinery is deficient or immature. Indeed, in patients with SCA, plasma ACTH levels do not correlate with the number of ACTH-positive cells of tumors, suggesting an altered secretion of an intact/biological active ACTH molecule [16].

Finally, another theory involves the role of tissue specific transcription regulators of POMC expression [20].

We here hypothesize that the mechanism involved in “the silencing” of SCA is multifactorial and involves disturbed transcription and processing of POMC, leading to an altered production and deficient secretion of modified ACTH molecules. Furthermore, there are clinical differences between SCA and FCA regarding their growth potential, and comparing their molecular profiles could reveal molecular targets that explain the increased aggressiveness of SCA.

The present study aims to investigate the differences between FCA and SCA by performing transcriptomics and proteomics analyses, followed by pathway discovery. The focus was to find the differences regarding hormone production and secretion (and possible medical treatment targets), and biomarkers of tumor growth and aggressiveness.

## 2. Results

### 2.1. Clinical Characteristics of Patients with FCA and SCA

Clinical characteristics of patients are presented in Appendix A. No significant difference was found in age, gender and Ki67-index between the two groups. Cortisol levels were higher in the FCA group (*p* = 0.007), as expected. Additionally, tumor size was significantly larger for the SCA group (*p* = 0.002) (Appendix A and Appendix A).

FCA expressed higher mRNA levels of POMC, TBX19 and PCSK1 than SCA (Figure 1A). However, there was an overlap in the expression of these corticotroph cell markers, several FCA tumors showing low levels, similar to those presented in SCA.

Gene expression of POMC was strongly associated with TBX19 (*R* = 0.74, *p* < 0.001) and PCSK1 (*R* = 0.48, *p* = 0.01) in FCA, but no significant association was observed in the SCA group (Figure 1B). TBX19 was also positively associated with PCSK1 in FCA (*R* = 0.59, *p* = 0.001), but not in SCA.

### 2.2. RNA-seq

Bioinformatics analyses revealed 631 differentially expressed genes (DEGs; fold change > 1.8 and *q* < 0.05). Of these, 286 DEGs were up-regulated and 345 down-regulated in FCA as compared to SCA. Heatmap analyses with hierarchical clustering and principal component analysis (PCA) analyses showed that samples grouped according to their tumor type in FCA and SCA without discrepancies (Figure 2A,C). Samples that clustered together presented a more similar gene expression profile.

### 2.3. Quantitative Label-Free Proteomics

Proteomic analysis identified 2782 proteins; among them, 170 were differentially expressed proteins (DEPs; fold change > 1.6 and *p* < 0.05) with 102 DEPs up-regulated and 68 DEPs were down-regulated in FCA as compared to SCA. Heatmap analyses with hierarchical clustering (Figure 2B) showed a more heterogeneous separation of the tumors. Several silent adenomas (SCA_10, SCA_9, SCA_8 and SCA_2) presented a protein expression profile more alike to FCA. No clinical particularities were observed for these four patients, although patient SCA_8 had a high plasma ACTH level, 14.7 pmol/L (normal range < 10.2), with normal serum cortisol, 477 nmol/L (normal range 112-502). and no clinical signs of hypercortisolemia. Furthermore, 8 of the 12 SCA tumors grouped together. The PCA plot (Figure 2D) showed more blunted separation of the two groups as compared to RNA-seq.

### 2.4. Common Molecules Presented in Transcriptomics and Proteomics Studies

Comparison of the differentially expressed genes and protein lists showed 38 differentially expressed molecules at both the mRNA and protein expression level (Table 1).

Several proteins belonging to collagen/cell adhesion/extracellular matrix (ECM) components were identified among the 38 molecules differentially expressed at both mRNA and protein level (Table 1). In addition, other cell adhesion genes were significantly increased only at mRNA level in FCA (COL1A1, COL4A3, COL4A4, COL20A1, COL28A1, LEPREL1, SERPINF1 and TIMP3) and SCA (COL2A1, COL11A1, SDC2, SPP1, FN1 and VTN) (Appendix A).

POMC and TBX19 were higher in the FCA group (Table 1). DKK3 was higher in the FCA group. In addition, other canonical WNT signaling activators (LGR5, RSPO3, and FZD5) were also higher expressed only at gene level in this group.

CRABP2, a protein known to potentiate effect of retinoids, was higher in SCA.

MAT2A showed discrepant regulation in FCA—down-regulated at the gene, but up-regulated at the protein level, perhaps explained by the fact that protein expression of this gene is regulated by post-transcriptional regulation [21].

Interestingly, the GNAS gene and protein were higher expressed in FCA, whereas the highest expressed molecule in the SCA group was NTS (neurotensin), a protein described in one study to be present in tumoral corticotroph cells [22].

### 2.5. Network Analysis in FCA and SCA

An interactive network representing the relations between all DEGs in both adenoma subtypes is presented online at https://alexeie.github.io/corticotroph-adenomas/. A simplified version is presented in Figure 3.

Figure 3 presents in a novel manner simultaneously the relationships between all the DEGs in both subgroups of adenomas. As shown, the relations between differentially expressed genes are complex, and drawing clear conclusions on the clinical significance is difficult. We therefore searched for biological processes and cellular components that were overrepresented in the data set using Panther (Appendix A). Based on our a priori hypothesis (i.e., FCA and SCA differ with regard to hormone production and secretion and growth potential), we further selected several biological processes and cellular components of relevance to the clinical implications and tumor phenotypes for detailed analyses (Appendix A) and created new network representations for FCA and SCA (Figure 4A,B). Protein folding in endoplasmic reticulum, regulation of transmembrane transport, regulation of cell migration, regulation of cell morphogenesis and differentiation, regulation of cell population proliferation, regulation of exocytosis, regulation of secretion and cell–cell adhesion were among the selected biological processed. With regard to the cellular component, the processes of importance for secretory cells (e.g., extracellular exosome, melanosome, endoplasmic reticulum lumen, chromaffin granule, cytoplasmic vesicle membrane and endocytic vesicle lumen) or cell adhesion/junction were selected (Appendix A).

Overrepresentation analyses showed that the FCA and SCA cohort both expressed genes related to biological processes that were expected from secretory endocrine cells (Appendix A). However, the expressed patterns of interactions were not identical. The SCA group expressed functional relations to protein and vesicle processing and to secretory vesicles, but the genes expressed bore few interactions between them. The FCA, however, expressed the same category of genes with a high degree of interactions.

Among the clusters of interactions overexpressed in FCA, one cluster related to hormone activity and endoplasmic reticulum protein processing and one cluster containing collagen/extra cellular matrix proteins were present (Figure 4A).

Among the genes related to hormone activity and endoplasmic reticulum protein processing, FCA showed up-regulation of genes involved in ribosome anchor (RRBP1), glycosylation and folding processes (RPN1, CALR, PDIA3, and UGGT2), protein recognition by luminal chaperones (HSPA5), and ER-associated degradation (HSPH1 and CRYAB). The only gene to be up-regulated in SCA was EIF2AK2, a protein known to phosphorylate translation initiation factor EIF2S1, which, in turn, inhibits protein synthesis. The difference with regard to endoplasmic reticulum protein processing genes was preserved with the exception of EIF2S1 when testing by RT-PCT in a larger cohort of cortiotroph adenomas [23] (Appendix A).

The ECM protein cluster up-regulated in FCA covered several collagen genes in addition to LAMA4, SERPINF1 and SPON2 (Figure 4A and Appendix A). The particular ECM expression profiles in FCA may be related to their “growth restriction” or even contribute to their active hormonal production.

CXCR4, a cancer stem cell marker in several tumors, inclusively in corticotroph adenomas [24], was higher expressed in FCA. However, Gene Ontology (GO)biological processes analysis (Appendix A) did not identify the stem cells marks category as one of the differentially expressed between FCA and SCA.

SCA expressed few clusters of biological activities (4 B,). However, a cluster of genes involved in vesicular transport/trafficking (STXBR5L, RAB3C, RAB8B and RIMS4) and another containing genes involved in cell-to-cell interaction (CACNG8, GRIK2, NLGN1, CADM1, and PVRL3) were present. Furthermore, few genes involved in NOTCH signaling (SGK1, CNTN1, and CNTN6) were also identified. SV2C, a protein coding gene which positively regulates vesicle fusion by maintaining the readily releasable pool of secretory vesicles [25], was also higher expressed in SCA.

The somatostatin receptor (SSR) status of different subgroups of adenomas is important, not the least with regard to the use of available medical treatments. There was no statistical difference in SSRs and dopamine receptor type 2 (DRD2) gene expression between functioning and silent corticotroph adenomas (Appendix A). SSR5 and SSR4 were not expressed in the SCA group.

The proteomics analysis identified fewer proteins compared to the number of genes identified by RNA-seq, therefore, the data are less abundant (Figure 5A,B and Appendix A). We performed network analysis by including all the DEPs.

Several evident protein clusters were identified in the FCA group (Figure 5A). One of them is composed of proteins known to be related to POMC, involving TBX19, PCSK1, GNAS, GFAP, and GALA. Other clusters were related to the extra cellular matrix (GPC1, DCN, COL15A1), neovascularization (VGF, TIPM2, QSOX1), calcium signaling (CACNA2D2, CAMk2B, CAMK2D), and glycolysis (PFKP, PFKFB2, IDH2). In addition, two key regulators of transforming growth factor beta, LTBP3 (also at gene level) and LTBP1, were also present.

Fewer protein clusters were identified in the SCA group (Figure 5B). Of note is the protein expression of PCSK1N, an inhibitor of PCSK1 in addition to two proteins involved in secretory granules (CHGB, SCG3) that were higher in SCA. Interestingly, several proteins known for their functions in the spliceosome were also present (SNRPE, SNRPG and CELF3). SSRs and DRD2 were not identified in the proteomics data set.

## 3. Discussion

We investigated, for the first time, the differences between functioning and silent corticotroph adenomas by performing RNA-seq and quantitative label-free mass spectrometry-based proteomics and analyzing network and pathway overexpression. The aim was to identify, first and foremost, the differences with regard to hormone production and secretion, as well as markers of growth and aggressive tumor behavior. The hierarchical clustering revealed two distinct groups in accordance with clinical and histological classification in the transcriptomic study, whereas in the proteomic study, there were four out of 12 silent tumors dispersed within the functioning group. Data visualization showed up-regulation of several genes involved in the protein processing in the endoplasmic reticulum (ER)-pathway in the FCA group. suggesting that an increased protein processing activity may lead to functioning corticotroph adenomas and raising the possibility for identification of a medical treatable target. Moreover, several molecules involved in cell adhesion were differentially regulated between FCA and SCA, indicating a different growth and aggressive potential. As expected, corticotroph cell markers were lower in SCA, and genes and proteins related to biological processes expected in secretory endocrine cells (e.g., vesicle-mediated transport and secretion) were present in both groups.

We present, in a large cohort of corticotroph adenoma, that gene expression of known corticotroph cell markers, POMC, TBX19, and PSCK1 was higher in the functioning group, in accordance with previous reports [26]. A strong association between POMC and TBX19 and PCSK1 expression was found in FCA but not in the SCA group. It is well known that TBX19 controls corticotroph cell identity and terminal differentiation, and inactivation of the TBX19 gene results in loss of POMC expression in corticotrophs [27], so the strong association between POMC and TBX19 was to be expected. Lower POMC and TBX19 expression in SCA supports the hypothesis that the presence of less differentiated cells in SCA is one of the mechanisms to explain diminished hormonal production and secretion. PCSK1 is involved in the initial cleaving of POMC in the anterior pituitary, the first step in acquiring a functional ACTH molecule [19]. Diminished expression of this enzyme in the SCA group, as presented in this study, may contribute to an impaired ACTH production in these tumors. Moreover, we demonstrate that protein level PCSK1N, a well described inhibitor of PCSK1-mediated processing of POMC [28], was significantly increased in SCA, suggesting an additional mechanism responsible for the impaired POMC processing in the silent adenomas. Of notice, a wide biological variability between the different patients was observed with regard to PCSK1 and PCSK2 genes and protein levels, potentially leading to a type-two error. In this sense, the omics studies showed that PCSK1 was significantly increased only at the protein expression level, whereas PCSK2 was significantly decreased only at the gene expression level in FCS.

Clustering of data revealed two distinct groups in the transcriptomic study in accordance with the tumors’ clinical phenotype. However, the proteomic study showed more variation with some silent tumors aggregating closer to the functioning counterparts. A lower number of identified proteins compared to the number of genes together with a greater degree of heterogeneity within the SCA may explain the more diffused delimitation observed at the protein level. Indeed, the network analyses also revealed fewer gene/protein clusters within the SCA group, reflecting a potential continuum of functionality, and therefore a broader variability, as previously suggested [5].

The presentation of the transcriptomics data as in Figure 3 is novel and original due to the simultaneously presentation of DEGs presented in both FCA and SCA, and added quantitative information (i.e., log2 fold change) in the visualization. In all the hitherto available visualization programs, the users have to manually separate up- and down-regulated genes and process them in separate analyses. Thereby, any result is only a partial representation of the underlying biological changes. In addition these figures ignore any underlying quantitative information [29].

Pathway enrichment analyses identified protein processing in the ER signaling pathway as being differentially regulated. The FCA group expressed up-regulation of several genes involved in ER protein folding, suggesting that they pose a higher processing activity compared to their silent counterparts. Indeed, a recent transcriptomic study identified that one of the main characteristics of the expression profile of corticotroph adenomas, per se, was overexpression of protein-targeting to the ER network [30]. Furthermore, it has been shown that unfolded/misfolded proteins accumulated due to the malfunctioning of the processing capacity in the ER leads to global translational repression in hypothalamic neurons and within the pituitary [31,32]. Our data suggest that SCA present a diminished ability for ER protein processing, and this may contribute to a decreased POMC gene and protein expression and therefore provide an explanation for their silence.

Another major, innovative finding of our study is the remarkable distinct ECM proteins and cell adhesion molecule profiles presented by the different adenoma types. It is obvious that the anterior pituitary gland comprises a multitude of cell types in addition to hormone specialized cells (e.g., folliculostellate cells, pituitary stem cells, and endothelial and connective tissue cells) [33,34]. All these cells may express and produce ECM components and cell adhesion molecules, although the contribution of each cell subtype is difficult to assess. Few studies have investigated different cell adhesion molecules and found them to be differentially expressed in SCA [35,36]. In our study, FCA presented up-regulation of several collagen genes, although SCA also overexpressed some other collagens. The differences were also evident at the protein level. The alterations seen between the two groups may be attributed to different cell populations or different expression of these molecules by similar cell types. Performing single-cell RNA-seq in FCA and SCA may give an answer to this question. In addition, further mechanistic studies to assess the role of these molecules will be needed to identify their importance in aggressiveness in SCA or perhaps growth restriction in FCA, or even in the regulation of hormone production as previously suggested [37].

The ECM plays a critical role in tumorigenesis and invasiveness/aggressiveness in several cancers, being a bridge between cell differentiation and proliferation, angiogenesis, and cell motility. The epithelial–mesenchymal transition process (EMT) has previously been studied in several subtypes of pituitary tumors and found to be of importance for both hormone production and aggressiveness [38,39,40,41]. Interestingly, in a recent multi-omics study, the silent corticotroph tumors presented low EMT, whereas USP8-wild-type functioning tumors showed and increased EMT, somehow challenging the classical imagistic phenotype of SCA as an aggressive tumor [42].

Our data showed that CRABP2, a member of the retinoic acid binding protein family, which transports retinoic acid to the nucleus and regulates its access to the nuclear retinoic acid receptors [43], was higher in the SCA group. Retinoic acids inhibit the transcription of POMC in corticotroph tumor cells, making the drug a candidate for treatment of Cushing disease [44,45]. Thus, the increased retinoic acid signaling in SCA may explain their reduced POMC levels and contribute to the silence.

Activation of calcium channels and intracellular calcium levels is regulated by CRH via CRH receptor 1 in the corticotrophs, leading to the release of secretory ACTH-containing vesicles [46,47]. Several calcium channel genes and proteins were up-regulated differentially in FCA and SCA. Functional characterization studies are needed to better understand their role on hormone regulation in corticotroph tumors.

GNAS genes and proteins, functioning as transducers in numerous signaling pathways controlled by G protein-coupled receptors, and known to mutate in several somatotroph adenomas [48], were higher expressed in FCA. Their importance in regard to hormone production in corticotroph adenomas, however, remains to be established.

LGALS3, a carbohydrate binding protein involved in apoptosis, innate immunity, cell adhesion and T-cell regulation, has been proposed as a molecular marker to differentiate between FCA and SCA [13,49,50]. Indeed, we found that LGALS3 was higher at the gene and protein level in FCA, and network analysis further identified MUC1, a protein involved in cell adhesion [51], as one of the interacting genes. Although activation of MUC1-mediated signaling in an autocrine/paracrine manner caused by ligation of LGALS3 promotes uncontrolled tumor cell malignancy [52], no role on hormone production/secretion has been described yet for this protein.

Galanine (GAL) was the protein with the highest fold of change in favor of FCA. However, the gene expression data did not display significantly different expression due to heterogeneous expression in SCA with some tumors expressing relatively high levels. Our data and previous reports suggest that GAL can be considered as a marker of corticotroph cell origin, although one cannot explain the large difference between the FCA and SCA groups, as no role in hormone production has been described [53,54].

In FCA, both GH and PRL showed higher expression at the gene and protein level, although the levels varied greatly between the samples. This was most probably due to normal pituitary tissue contamination of the small functioning corticotroph tumors, as GH is one of the highest expressed genes in the normal pituitary [30,34,55,56]. However, when the data were analyzed after the FCA samples with high expression GH and PRL were removed, the results and their interpretation were similar.

There was no clear evidence of the presence of consistent differences in the activation of classical developmental signaling pathways (i.e., TGF-beta signaling, WNT signaling, Hippo signaling, Notch signaling) between the groups. However, we identified few markers to be significantly different. These signaling pathways have often several layers of regulation, and activation often implies phosphorylation of key proteins, so one should be cautious when interpreting gene data alone.

DKK3, found to be increased at the gene and protein level in FCA, is considered to be a tumor suppressor involved in several cancers [57] and, most interestingly, described as required for maintaining the integrity of secretory vesicles in mouse adrenal glands [58]. Further studies with regard to this marker might reveal its role in modulating both hormone production and tumor growth restriction in corticotroph adenomas.

This study has some limitations. Pre-operative MR pictures were not available for all patients, thus limiting the information of invasiveness of the tumors. However, the main aim of the study was to explore the differences between functioning and non-functioning corticotroph adenomas with regard to their transcriptome and proteome. Thus, invasiveness was not considered as one of our criteria. As for other studies on SCA [8], preclinical evaluation of cortisol rhythm was not available for this group. All the SCA patients were clinically evaluated pre-operatively at our tertiary referral center, and no clinical signs of cortisol hypersecretion were documented. Of note, and a strength of this study, is that the IHC staining for TBX19 and ACTH was performed in a research setting in one run for all tumors, using a specific antibody to detect transcription factor TBX19 [59]. Lastly, the mutational status of USP8 was unknown. A recent pangenomic classification of pituitary neuroendocrine tumors divided the corticotroph functioning tumors into two classes: the USP8-mutated type with overt secretion, and the USP8-wild-type with increased invasiveness and increased epithelial–mesenchymal transition [42]. Our FCA group contains potentially a blending of USP8-wild-type and mutated tumors, potentially diminishing the power of identifying differences. Moreover, the transcriptomics analysis revealed that all the FCA grouped together, with no obvious separation between them.

## 4. Material and Methods

### 4.1. Sample Selection

The samples included in this study were selected from a larger cohort of pituitary adenomas [41,60,61] (FCA, *n*  =  48, and NFPA, *n* = 255) based on positive IHC staining for ACTH and TBX19, expression of POMC and TBX19 by real-time quantitative PCR (RT-qPCR) and sufficient RNA quality. RT-qPCR for POMC, TBX19 and PCSK1 was performed in an extended cohort of adenomas (FCA *n* = 27, SCA *n* = 10) in one run. RNA-seq was performed on 6 FCA and 6 SCA, whereas proteomics was performed on 12 FCA and 12 SCA (Figure 6). The amount of protein was insufficient in two FCA, so only four FCA and six SCA were used for both RNA-seq and proteomics studies.

The clinical characteristics of patients included in the RNA-seq and proteomics studies are presented in Appendix A. The patients diagnosed with FCA had signs and symptoms of endogenous hypercortisolism, and the clinical diagnosis of Cushing’s disease was confirmed in all patients. SCA was diagnosed because of tumor symptoms such as visual field defect, headache and/or hormone deficit symptoms (i.e., amenorrhea) and did not present signs and symptoms of endogenous hypercortisolism.

Written informed consent was obtained from all patients. The study was approved by the regional ethics committee (REK no: 2014/1680, REK no: 2014/635 and REK no: 2020/22301) and hospital authority. Data were analyzed using IBM SPSS statistics version 26.0 (SPSS, Chicago, IL, USA).

### 4.2. RNA Isolation and Reverse Transcription

Tissue was homogenized in Trizol (Invitrogen, Carlsbad, CA, USA), and RNA was purified using QIAGEN miRNeasy Mini Kit (Qiagen, Valencia, CA, USA), according to the manufacturer’s instructions, including the step for removal of genomic DNA. RNA integrity was determined using the Agilent 2100 Bioanalyzer (Agilent Technologies, Santa Clara, CA, USA), and concentrations were measured by optical density OD readings on a Nanodrop ND-1000 Spectrophotometer (Nanodrop Technologies, Wilmington, DE). All RNA integrity numbers (RIN) were above 7, indicating a high RNA quality [62].

Reverse transcription was performed using a High Capacity cDNA Reverse Transcription Kit (Applied Biosystems, Foster City, CA, USA) in a Labnet MultiGene Gradient Thermal Cycler (Labnet International Inc, Edison, NJ, USA) according to the manufacturer’s protocol using a total of 1 μg RNA. After the reaction, the cDNA was diluted to a ratio of 1:10.

### 4.3. RT-qPCR

RT-qPCR was performed in the ABI 7900 (Applied Biosystems, Foster City, CA, USA). Applied Biosystems™ Power™ SYBR™ Green Master Mix and samples were dispensed in the corresponding wells by an automated pipetting system (epMotion^®^ 5070 CB, Hamburg, Germany).

Primer pairs were found either in the literature or in the primer bank (Harward medical school C., 2006) and are presented in Appendix A primers. All primers were tested for specificity using BLAST analysis (NCBI). To avoid false positive amplification due to DNA contamination, the genomic DNA sequence (downloaded from Ensemble Data Base www.ensembl.org), was used to check exon–intron borders by matching the primers to their location. The amplification efficiency and correlation coefficients for these primers (Sigma Aldrich, St. Louis, MO, US) were obtained from the slope of the standard curves. All RT-qPCR experiments were in accordance with the Minimum Information for Publication of Quantitative Real-Time PCR Experiments (MIQE) guidelines. Data were adjusted to geometrical mean of two housekeeping genes (GAPDH and ALAS1) as previously validated [63].

### 4.4. RNA-Seq

RNA-seq was performed using 2 μg RNA for each sample (*n* = 12) at the Norwegian Sequencing Centre, Oslo, Norway. Twelve RNA-seq libraries with unique indexes were prepared using TruSeq stranded mRNA library prep kit (Illumina, San Diego, CA, USA) by following the manufacturer’s instructions. Two batches of six libraries were pooled together, and 150 nt paired-end sequencing was performed on one lane of HiSeq 4000 (Illumina) for each batch. RTA v1.18.66.3 was used for base calling and was further processed using bc2fastq v2.17.1.14 to demultiplex and generate fastq data based on the indexes used during library preparation.

### 4.5. Pre-Processing and Cleanup

Low-quality reads and adaptors were removed using Trimmomatic v0.33 [64] with recommended parameters. BBMap v34.56 (http://sourceforge.net/projects/bbmap) was used to remove reads aligning to PhiX (RefSeq: NC_001422.1), which was added as a spike-in during sequencing.

### 4.6. Transcriptome Alignment

Cleaned data were aligned against the Human ensemble GRCh38 (p10, release 90) genome and the transcriptome using Tophat2 v2.0.13 [65], using ‘--library-type fr-firststrand --no-mixed --no-novel-juncs --transcriptome-index’ as parameters. Library size was estimated using the picard v1.112 (http://broadinstitute.github.io/picard/) CollectInsertSizeMetrics tool after aligning the first one million reads using bowtie v2.2.3 [66] to human ensemble GRCh38 cDNA sequences, and the output was also provided as parameters for tophat2 alignment. Cuffdiff v2.2.1 [67] pipeline was used to calculate the differential expression of the known genes described in ensemble General Feature Format (GTF) and CummeRbund v2.14 [68]. R package was used to visualize expression data, and custom scripts were used to create tables and graphs. Hierarchical clustering was performed using Pearson correlation with complete linkage in TM4 MeV (http://mev.tm4.org/).

### 4.7. Quantitative Label-Free Mass Spectrometry-Based Proteomic Analysis

Tissue samples stored in TRIzol™ reagent (Thermo Fisher Scientific, Invitrogen, Carlsbad, CA, USA) were homogenized according to the manufacturer’s protocol. RNA supernatant was removed, ethanol was added for DNA precipitation, and the resulting protein pellet was utilized for proteomic analysis. The proteins were dissolved in 8M urea in 50 mM NH_4_HCO_3_, reduced and alkylated, and digested into peptides with trypsin (Promega, Madison, Wisconsin, USA). The resulting peptides were desalted and concentrated before mass spectrometry (MS) by the STAGE-TIP method using a C18 resin disk (3M Empore, St. Paul, MN, USA). Each peptide mixture was analyzed by a nEASY-LC coupled to a QExactive Plus (ThermoElectron, Bremen, Germany) with an EASY Spray PepMap^®^RSLC column (C18, Thermo Fisher Scientific, Carlsbad, CA, USA).

The resulting MS raw files were submitted to the MaxQuant software version 1.6.1.0 for protein identification and label-free quantification. Carbamidomethyl (C) was set as a fixed modification, and acetyl (protein N-term), carbamyl (N-term) and oxidation (M) were set as variable modifications. A first search peptide tolerance of 20 ppm and a main search error 4.5 ppm were used. Trypsin without the proline restriction enzyme option was used, with two miscleavages allowed. The minimal unique + razor peptide number was set to 1, and the allowed false discovery rate (FDR) was 0.01 (1%) for peptide and protein identification. Label-free quantitation was employed with default settings. The Uniprot database with “human” entries (September 2018) was used for the database searches.

Known contaminants as provided by MaxQuant and identified in the samples were excluded from further analysis. Perseus version 1.6.1.3 was used for further analysis of MaxQuant data: label-free quantitation (LFQ) intensity values were log10 transformed, a minimum of 50% valid values in at least one group was required, and missing values were imputed from the low end of normal distribution. Student’s *T*-test between the groups (*p* < 0.05) was performed to find the significant DEPs.

### 4.8. Pathways

For comparison purposes, protein level values were changed from log10 to log2 in order to facilitate comparison with RNA-seq data. Proteins with a log2 fold change higher than 1 (i.e., a fold change of 2) and genes with a log2 fold change higher than 0.92 (i.e., a fold of change 1.8) were considered for network and pathway analysis.

After a Student’s *T*-test was performed, the resulting set of differentially expressed genes (DEGs)/differentially expressed proteins (DEPs) were utilized in an overrepresentation test in Panther-db with Fischer’s exact test controlling for false positives by Benjamini Hochberg’s false discovery rate (FDR). For DEGs, the complex list of pathways was also further curated by eliminating pathways that were not considered informative to the pituitary tumor phenotype or involved in hormone production/secretion and tumor growth. DEPs were presented in the network without any selection due to the fact that the proteomic data included only 170 DEPs and the figures were less complex.

The interactions between the detected gene products’ and proteins’ remaining biological pathways were investigated by plotting the corresponding genes’ protein interactions in the String-db web interface with medium confidence while hiding disconnected nodes.

To assess if the DEPs displayed any network connectivity (and thus pointing to biological pathways of interest), the online String database and Panther database were employed to visualize and assess the presence of protein interactions [69,70].

Overrepresentation analysis was performed with Panther-db using Homo sapiens as a reference genome. Two-log fold enrichment of the overrepresentation test ranged from 1.36 to 4.77 and from 0.96 to 6.64 for the selected genes and proteins, respectively.

### 4.9. Heatmap and PCA

A heatmap was generated for transcriptomic and proteomic data by use of R packages Plotly and Heatmaply [71,72]. The heatmap displays significantly expressed transcripts and proteins on a Z-scored color scale that indicates the number of standard deviations that each variable deviates from the variable mean, and the branches are colored according to which second-level branch they belong to.

Principal component analysis (PCA) was performed on both data sets, on complete data without any selection. PCA plots for RNA-seq data and proteomic data are represented with 41% and 48% of total variance, respectively.

## 5. Conclusions

Distinct clinical aspects of FCA and SCA may be explained by their different repertoires of activated signaling pathways, namely, promoting growth in SCA and protein processing in FCA and their specific patterns of cell adhesion molecules. Further in vitro functional studies should be performed in order to identify possible medical targets and to properly understand the involved mechanisms.

## Figures and Tables

**Figure 1 cancers-12-02980-f001:**
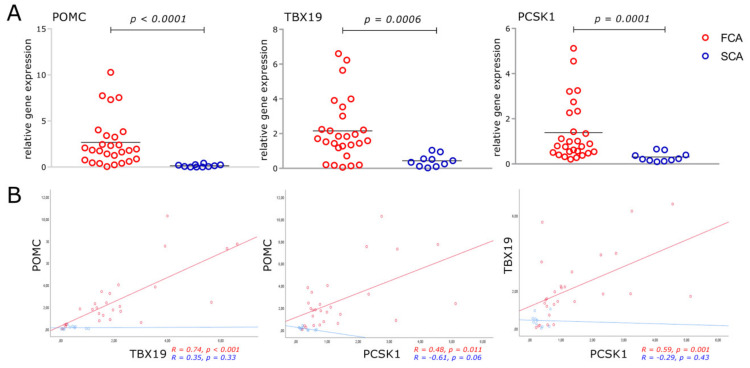
mRNA expression of POMC, TBX19 and PCSK1 in functioning (FCA) and silent (SCA) corticotroph adenomas. Data are presented as a scatter dot plot and mean (**A**) and correlation plots with linear regression lines (**B**). A Mann–Whitney nonparametric test and Spearman correlation were performed; FCA *n* = 27; SCA *n* = 10;

**Figure 2 cancers-12-02980-f002:**
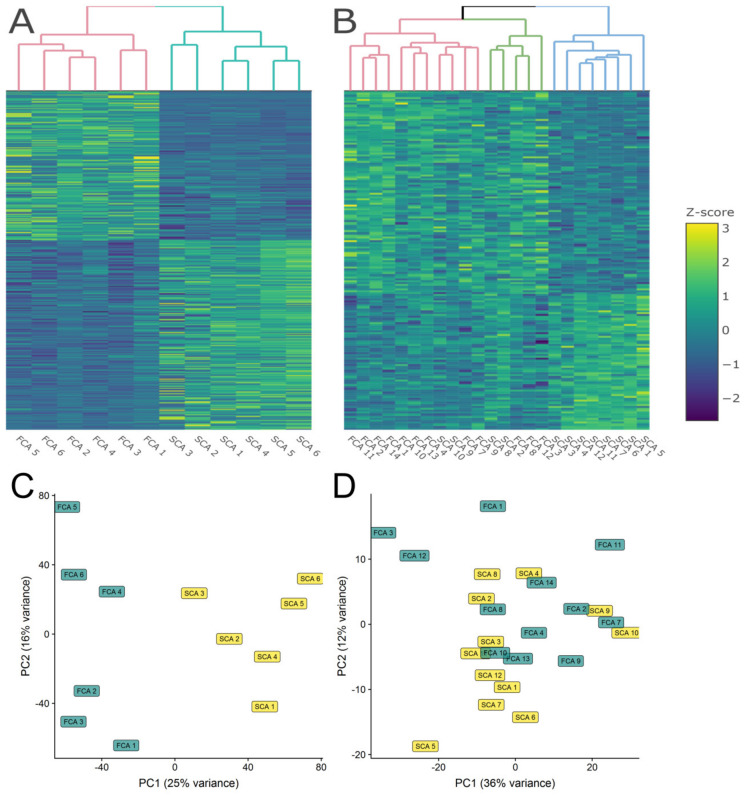
Heatmap (**A**,**B**) and principal component analysis (PCA; **C**,**D**) profiling of RNA-seq differentially expressed genes (DEGs; **A**,**C**) and proteomic differentially expressed proteins (DEPs; **B**,**D**) in functioning (FCA) and silent (SCA) corticotroph adenomas.

**Figure 3 cancers-12-02980-f003:**
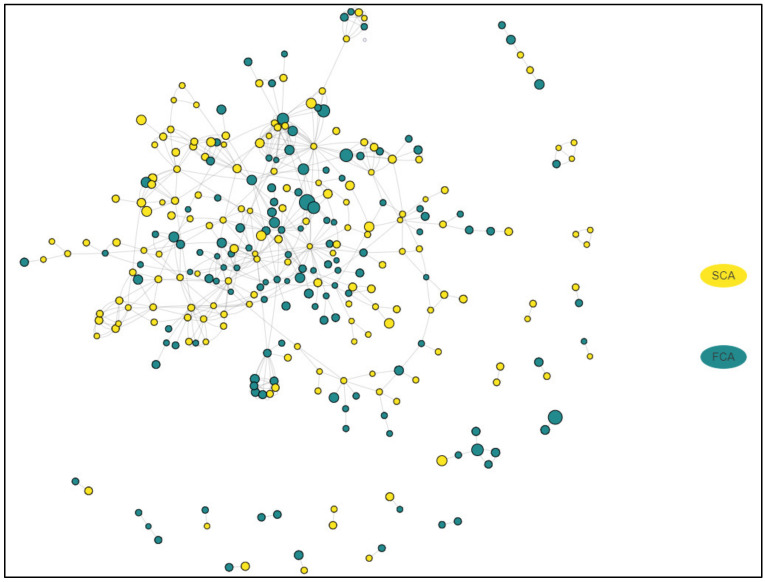
Interactive network representing the relations between all DEGs in functioning (FCA) and silent (SCA) corticotroph adenomas. The figure shows a representation of the complexity of the results and the interactions that are present. An interactive version of this figure is presented at https://alexeie.github.io/corticotroph-adenomas/. Detailed information on all the key interactions in the dataset, including the involved genes and the known relationships between them, can be accessed via the link. Genes higher expressed in FCA group are green; genes higher expressed in SCA are yellow. The size of the round circles is proportional to the log2 fold change.

**Figure 4 cancers-12-02980-f004:**
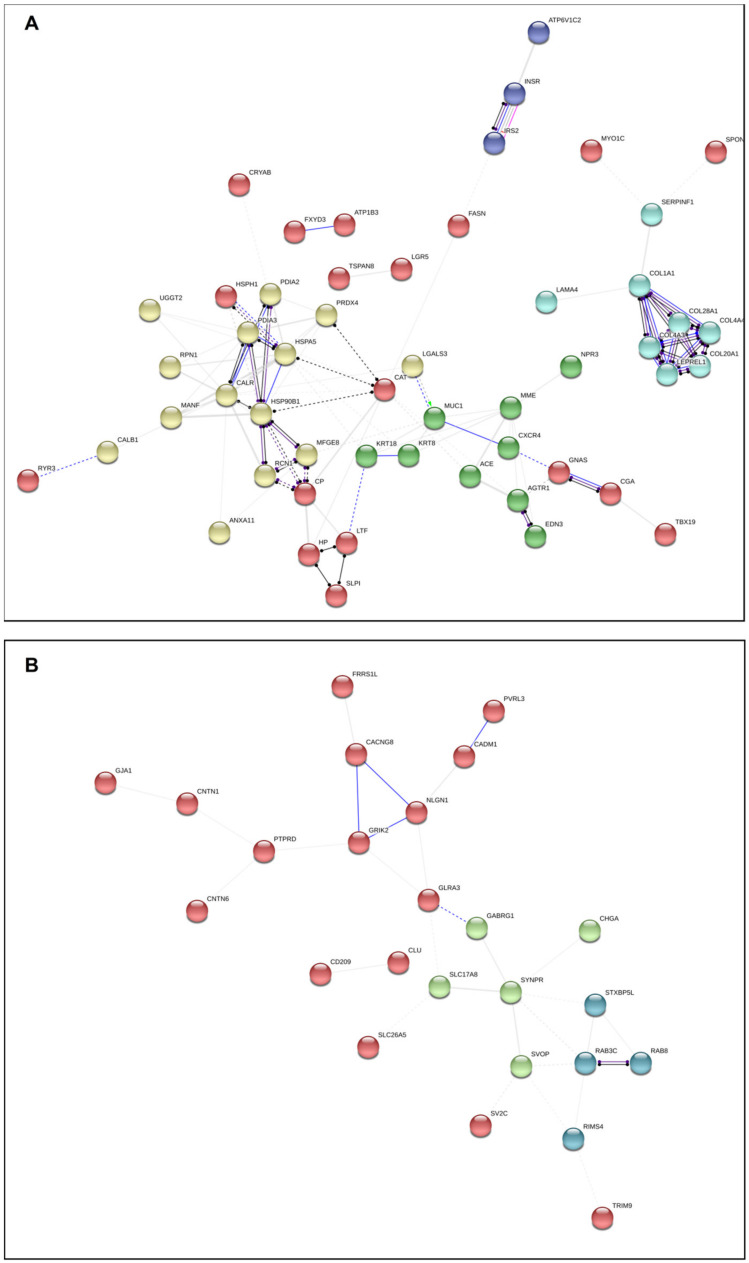
Network analysis presenting DEG interactions in functioning (FCA) (**A**) and silent (SCA) (**B**) corticotroph adenomas based on selected distinct biological processes and cellular components overrepresented in FCA and SCA. GO processes representative of the pituitary tumor phenotype or involved in hormone production/secretion and tumor growth were selected to be presented in the figure. For a comprehensive list of all the significant processes in the overrepresentation test and the selected ones, please see Appendix A (biological_process_SEQ, cellular_component_SEQ, GO selected_SEQ and GO matches_SEQ) and the online figure for FCA at https://version-11-0.string-db.org/cgi/network.pl?networkId=6sMwLJjOC4ua and for SCA at https://version-11-0.string-db.org/cgi/network.pl?networkId=pdN7O6hmQ5G3. The color of the circles indicates the result of a K-means algorithm for identifying clusters of genes according to their interaction partners.

**Figure 5 cancers-12-02980-f005:**
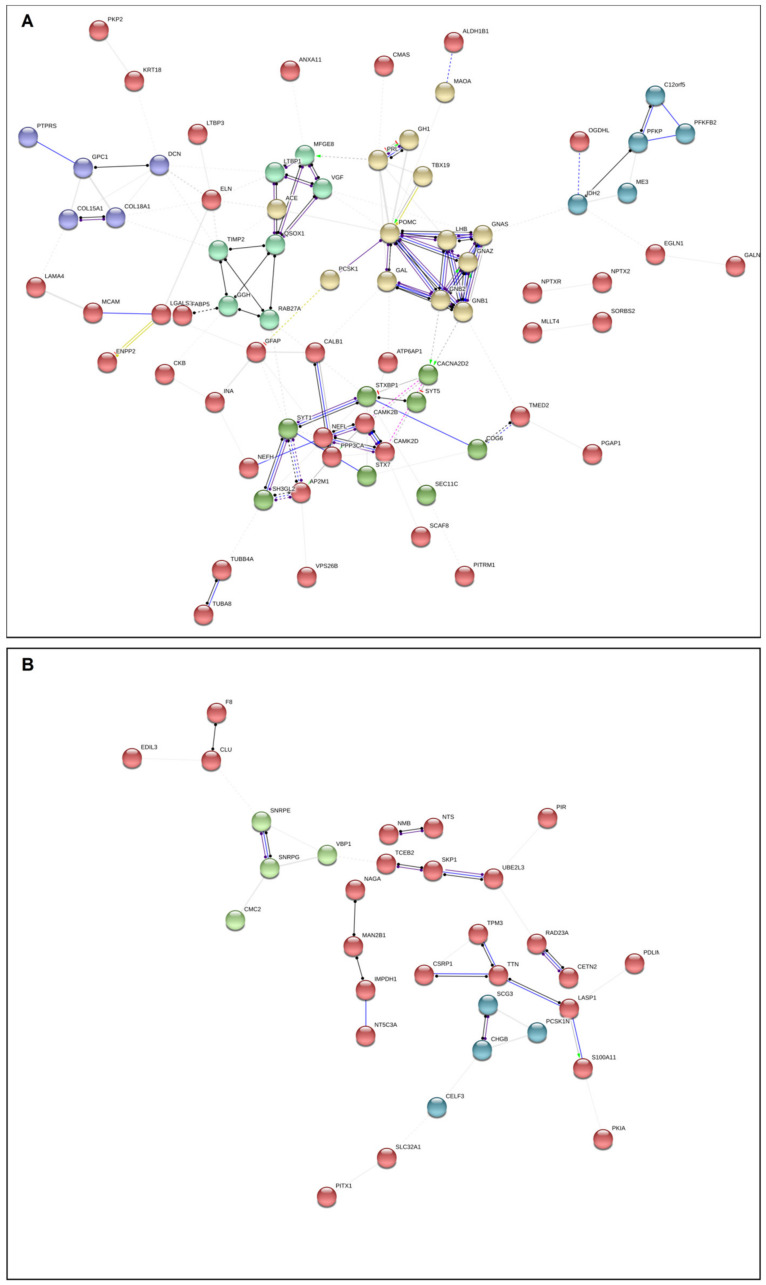
Network analysis presenting all DEP interactions in functioning (FCA) (**A**) and silent (SCA) (**B**) corticotroph adenomas. See Appendix A and online figures for FCA https://version-11-0.string-db.org/cgi/network.pl?networkId=nBtGbCu86iDj and for SCA https://version-11-0.string-db.org/cgi/network.pl?networkId=eC3B7NayyvdC for more detailed information. The color of the circles indicates the result of a K-means algorithm for identifying clusters of genes according to their interaction partners.

**Figure 6 cancers-12-02980-f006:**
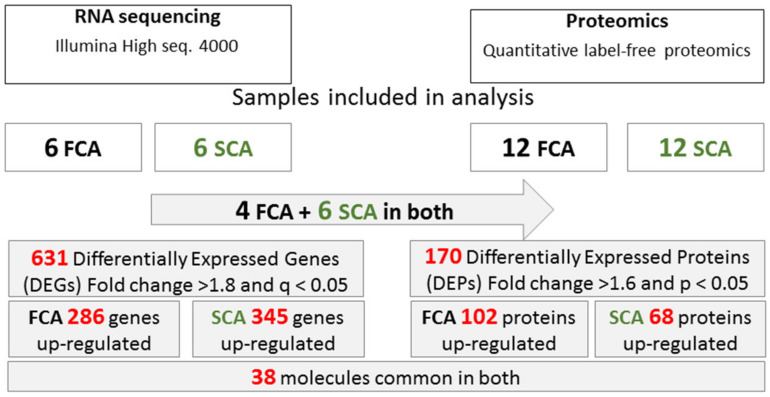
Study design.

**Table 1 cancers-12-02980-t001:** Common molecules presented in transcriptomics and proteomics studies. Thirty-eight molecules were differentially expressed at both mRNA and protein level in functioning (FCA) and silent (SCA) corticotroph adenomas. Molecules up-regulated in FCA are presented with positive fold of change values.

Gene Short Name	Gene Description	Cell Adhesion	RNA-seq Fold Change	Proteomics Fold Change
GH1	growth hormone 1		1982.1	18.1
PRL	prolactin		176.1	19.8
POMC	proopiomelanocortin		33.3	9.7
LGALS3	galectin 3	X	22.8	7.7
ELN	elastin		14.3	1.9
KRT18	keratin 18	X	13.1	9.6
CACNA2D2	calcium voltage-gated channel auxiliary subunit alpha2delta 2		8.7	3.5
ENPP2	ectonucleotide pyrophosphatase/phosphodiesterase 2	X	6.0	5.2
DKK3	dickkopf WNT signaling pathway inhibitor 3		5.4	7.6
ANXA11	annexin A11		5.1	3.3
LAMA4	laminin subunit alpha 4	X	4.9	2.6
MAOA	monoamine oxidase A		4.5	2.9
MFGE8	milk fat globule-EGF factor 8 protein		4.5	9.0
ACE	angiotensin I converting enzyme		4.4	2.7
DCN	decorin	X	4.2	6.6
CALB1	calbindin 1		3.9	10.2
TBX19	T-box 19		3.8	4.9
SORBS2	sorbin and SH3 domain containing 2		3.3	3.4
KCTD12	potassium channel tetramerization domain containing 12		2.7	2.5
APMAP	adipocyte plasma membrane associated protein		2.6	1.9
LTBP3	latent transforming growth factor beta binding protein 3		2.5	4.1
GNAS	GNAS complex locus		2.3	1.9
MAT2A *	methionine adenosyltransferase 2A		-2.5	1.7
LASP1	LIM and SH3 protein 1	X	-2.5	−2.4
CAB39L	calcium binding protein 39 like		-2.7	−3.1
CSRP1	cysteine and glycine rich protein 1		-3.0	−2.9
QPCT	glutaminyl-peptide cyclotransferase		-3.1	−3.5
CLU	clusterin	X	-3.9	−3.5
S100A11	S100 calcium binding protein A11	X	-4.2	−5.7
DBI	diazepam binding inhibitor, acyl-CoA binding protein		-4.5	−4.2
PDLIM1	PDZ and LIM domain 1	X	-5.2	−5.3
FAM107B	family with sequence similarity 107 member B		-5.9	−2.8
SLC32A1	solute carrier family 32 member 1		-7.2	−4.5
GPC4	glypican 4	X	-8.6	−3.0
EDIL3	EGF like repeats and discoidin domains 3	X	-8.7	−6.7
CRABP2	cellular retinoic acid binding protein 2		-9.9	−8.7
LAMC2	laminin subunit gamma 2	X	-12.9	−9.6
NTS	neurotensin		-45.6	−46.6

* MAT2A was down-regulated at gene expression but up-regulated at protein expression in FCA as compared to SCA; X; molecules involved in cell adhesion.

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
