# Peer review of "Distinct Pattern of Endoplasmic Reticulum Protein Processing and Extracellular Matrix Proteins in Functioning and Silent Corticotroph Pituitary Adenomas"

_cancers, 2020, doi:10.3390/cancers12102980_

Round 1
Reviewer 1 Report
The authors tried to discover the key molecules that contribute to making clinically different characteristics between functioning (FCA) and silent corticotroph adenoma (SCA). Among the hundreds of differentially expressed molecules, the authors eventually identified some statistically significant pathways and biological functions through over-representative analysis (ORA) and investigate molecular network constructed of DEGs (DEPs in proteomics). I think the multi-omics approach in both FCA and SCA is new and the authors' attempts are desirable. However, I am very suspicious whether this manuscript could be published, as the results of bioinformatics are somewhat preliminary and the conclusion is so typical that anyone might expect. Moreover, since RNA-seq and MS-based proteomics are not integrated, an analysis that can lead to interesting results has not been attempted, thus the description of the results that are deemed to be not much important is repeated. A substantial portion of the study should be revised before re-submission. The major issues of this paper are as follows.
Major points #1
In the Introduction section, the description of a previous study in lines 66-67 is over-abbreviated. Please provide detailed information about it. Besides, I would like the authors to investigate and summarize the previous omics studies on FCA, SCA, and PitNETs performed by other research teams. In addition, to present the limitations of previous studies and describe the strengths of this study over these studies will make this paper more competitive.
Major points #2
I suggest the authors to provide the MR image or IHC image (with FCA- or SCA-specific markers stained) or ACTH blood concentration (of all patients) as supplementary data to ensure that the tissue samples used for the omics analyses are representative of the FCA and SCA. Patient cohort information, provided only by age and gender, cannot solely guarantee that the two groups of cancer tissue samples used in this study are clinically distinct.
Major points #3
In the Results section, I wonder if the heatmaps in Figure 2 are drawn using only the expressions of DEP. If so, the clusters mentioned by the authors are not the result of “unsupervised clustering”. If the authors set DEP conditions and performed clustering using these selected molecules, it is already supervised. Please correct the terms in other sections such as the Discussion section.
In addition, I suggest the authors actually perform unsupervised clustering using all molecules (RNAs in RNA-seq and proteins in proteomics). Then, the authors can explain the new cluster located between FCA and SCA in the PCA plot. If this cluster is remained in unsupervised clustering, how about performing a bioinformatics analysis on it? The four samples that belong to this cluster must be of interest to the scientific community.
Major points #4
In the subsection “2.3. Integrative transcriptomics and proteomics” of the Result section, there is no integrative analysis actually performed. What is the reason that the authors named this subsection like this? In addition, why is the molecular interaction network constructed in the next subsection? Why did the network showed in figure 3 is transcriptome-proteome-combined while subsequent ORA are performed separately for each molecule types? I think that both figure 3 and the network in the URL (https://alexeie.github.io/corticotroph-adenomas/) give no information. If the authors have thought to perform the integrative omics analysis, please review the below articles.
https://www.ncbi.nlm.nih.gov/pmc/articles/PMC3637682/
https://pubs.acs.org/doi/10.1021/acs.jproteome.8b00778
In summary, what are the important biological processes and proteins in this molecular interaction network?
Major points #5
In Figure 4, the molecular interaction network is drawn again with the DEGs belonging to several selected BPs as the authors mentioned in line 162-163. It is obvious that the genes associated with identical or similar BPs must form tight associations in a molecular interaction network when the authors used gene members selected in advance. So, what is the new information this network provides? The current description in line 170-180 and 193-205 is not different from the simple summarization of selected ORA results performed using PANTHER-DB.
The analysis of the protein network represented by Figure 5 also has the same problem as in the above analysis. In addition, line 212-233 tends to repeat the same things.
Last, it would be better to provide the molecules associated with the selected BPs in the table. And, it may be helpful to analyze the proteins or genes that act as a hub in the above networks.
Major points #6
Some contradictions within the paper need to be corrected or discussed. I wonder if the expression of PSCK1, a corticotroph cell marker, showed significant differential expression in the proteomics dataset because it is not included in Table 1 which listed the overlap of DEG and DEP. However, in line 199-200, the authors described as it is increased in proteomic data. Please clarify this. If the expression of proteins, both PSCK1 and PSCK2, are not significantly altered in the proteomics data, please discuss the reason.
The authors mentioned in lines 132-133 that Wnt signaling is more activated in FCA than SCA. However, in the Introduction section, as the authors stated, the more aggressive one is SCA. An explanation for this inconsistency is required. Likewise, the BP “neovascularization” might be a hallmark of SCA because it is aggressive cancer. Why is this BP enriched only in FCA? Conversely, the BP “hormone processing and secretion” seems to be a characteristic of FCA. However, this BP term is enriched in FCA. If this BP term is related to other hormones except for ACTH, please discuss them.
Major points #7
The non-MS methods (e.g. Western blot, ELISA, IHC, RT-PCR, etc.) are needed to support the omics results. The authors suggested that the genes CRABP, LGALS3, GAL as important proteins in the Discussion section. If there are additional experimental results for the expressions of these genes, it will be a concrete basis for the authors' suggestions.
Minor points #1
I suggest the authors to supplement the contents in lines 122-128 with more details. Simply listing collagen/cell adhesion/extracellular matrix (ECM)-related genes or proteins does not give the readers any information.
Minor points #2
In lines 134-143, a brief description of each significant protein is written. However, the association between these proteins and PitNET is not described. Please complement detailed information.
Minor points #3
There are many paragraphs (or sentences) in this paper that are cut off in halfway of explaining something. For example, the paragraph covered line 187-188 has an interesting introduction like “The ECM proteins cluster covered several collagen genes in addition to LAMA4 and SERPINF1 and SPON2 (Figure 4 A and Suppl.Excel.File_GO_matches FCA_SEQ).”, but the authors instantly stop explain specific things. This significantly interferes with reading. The overall revision of writing is needed.
Author Response
Point-by-point response to the Reviewers’ comments
We sincerely thank the Reviewers for their constructive comments, which we found very helpful towards improving the quality of our manuscript. Accordingly, specific changes have been made in the manuscript, based on these comments, as it is described in detail below in a point-by-point response to the reviewers. Changes within the manuscript are highlight in yellow.
RESPONSES TO Reviewer #1
The authors tried to discover the key molecules that contribute to making clinically different characteristics between functioning (FCA) and silent corticotroph adenoma (SCA). Among the hundreds of differentially expressed molecules, the authors eventually identified some statistically significant pathways and biological functions through over-representative analysis (ORA) and investigate molecular network constructed of DEGs (DEPs in proteomics). I think the multi-omics approach in both FCA and SCA is new and the authors' attempts are desirable. However, I am very suspicious whether this manuscript could be published, as the results of bioinformatics are somewhat preliminary and the conclusion is so typical that anyone might expect. Moreover, since RNA-seq and MS-based proteomics are not integrated, an analysis that can lead to interesting results has not been attempted, thus the description of the results that are deemed to be not much important is repeated. A substantial portion of the study should be revised before re-submission. The major issues of this paper are as follows.
Major points #1
In the Introduction section, the description of a previous study in lines 66-67 is over-abbreviated. Please provide detailed information about it. Besides, I would like the authors to investigate and summarize the previous omics studies on FCA, SCA, and PitNETs performed by other research teams. In addition, to present the limitations of previous studies and describe the strengths of this study over these studies will make this paper more competitive.
As the reviewer suggests we have now supplied with more information the study referred in lines 66-67. Regarding previous omnics studies comparing FCA and SCA: We have already discussed extensively in the manuscript the results published in the only RNA seq study that includes corticotroph functioning and silent tumors ref.43 doi:10.1016/j.ccell.2019.11.002). Our study is unique since it describes, to the best of our knowlage, for the first time the molecular differences at gene and protein level in two cohorts functioning- and slilent- corticotroph tumors. For the sake of space and clarity we did not included information on the previous omnics studies performed in other PitNET subtypes.
Major points #2
I suggest the authors to provide the MR image or IHC image (with FCA- or SCA-specific markers stained) or ACTH blood concentration (of all patients) as supplementary data to ensure that the tissue samples used for the omics analyses are representative of the FCA and SCA. Patient cohort information, provided only by age and gender, cannot solely guarantee that the two groups of cancer tissue samples used in this study are clinically distinct.
Thank you for your suggestion. We have considered it and have added a supplementary figure presenting the typical MRI characteristics of FCA and SCA. In addition we have supplemented the Suppl. File. Table S1 and the results section with available data on Ki67, ACTH and Cortisol levels in all the included patients.
Major points #3
In the Results section, I wonder if the heatmaps in Figure 2 are drawn using only the expressions of DEP. If so, the clusters mentioned by the authors are not the result of “unsupervised clustering”. If the authors set DEP conditions and performed clustering using these selected molecules, it is already supervised. Please correct the terms in other sections such as the Discussion section.
As described in the Material and methods section (l.467-468) the heatmaps presented in the figure 2 display significantly differentially expressed transcripts (A) and proteins (B).
In addition, I suggest the authors actually perform unsupervised clustering using all molecules (RNAs in RNA-seq and proteins in proteomics). Then, the authors can explain the new cluster located between FCA and SCA in the PCA plot. If this cluster is remained in unsupervised clustering, how about performing a bioinformatics analysis on it? The four samples that belong to this cluster must be of interest to the scientific community.
Major points #4
In the subsection “2.3. Integrative transcriptomics and proteomics” of the Result section, there is no integrative analysis actually performed. What is the reason that the authors named this subsection like this? If the authors have thought to perform the integrative omics analysis, please review the below articles.
https://www.ncbi.nlm.nih.gov/pmc/articles/PMC3637682/
https://pubs.acs.org/doi/10.1021/acs.jproteome.8b00778
As we already acknowledged in the first paragraph, we do agree with the reviewer that no “integrated” analysis has been performed. We believe that it might be difficult to come up with any meaningful conclusion by performing “integrated” analysis on just a few patients given that they are biologically different. The intention of this manuscript was not to perform the “integration” as mentioned in the two articles recommended by the reviewer but to compare the RNAs-seq and proteomics dataset and find the common molecules. Accordingly we changed the word “Integrative” with “comparative analysis on differentially expressed genes and protein lists”
In addition, why is the molecular interaction network constructed in the next subsection? Why did the network showed in figure 3 is transcriptome-proteome-combined while subsequent ORA are performed separately for each molecule types? I think that both figure 3 and the network in the URL (https://alexeie.github.io/corticotroph-adenomas/) give no information.
We thank for pointing out the ambiguous description of the figure. Figure 3 presents all the differentially expressed genes (n=631) in functioning (FCA) and silent (SCA) corticotroph PitNETs. We have now supplied the manuscript with the following information: The figure 3 shows a representation of the complexity of the results and the interactions that are present between the differentially expressed genes. An interactive version of this figure is presented at https://alexeie.github.io/corticotroph-adenomas/. A closer look at all the key interactions in the dataset can be achieved via the link. Genes higher expressed in FCA group are green; genes higher expressed in SCA are yellow. The size of the round circles is proportional to the fold change.
In summary, what are the important biological processes and proteins in this molecular interaction network?
Actually, by plotting all the differentially expressed genes in a network, it is possible to understand and visualise the complexity of the data better, and easily search for the known and perhaps discover new interactions between the DEG. The overall aim was to understand the molecular differences between these two tumor subtypes. Figure 3 shows just the DEG, no information on DEP is presented. The important/ statistical significant processes at both gene and protein levels are described in the suppl. files (please see Suppl.Excel.File_biological_processes and Suppl.Excel.File_cellular_components).
Major points #5
In Figure 4, the molecular interaction network is drawn again with the DEGs belonging to several selected BPs as the authors mentioned in line 162-163. It is obvious that the genes associated with identical or similar BPs must form tight associations in a molecular interaction network when the authors used gene members selected in advance. So, what is the new information this network provides? The current description in line 170-180 and 193-205 is not different from the simple summarization of selected ORA results performed using PANTHER-DB.
The analysis of the protein network represented by Figure 5 also has the same problem as in the above analysis. In addition, line 212-233 tends to repeat the same things.
We have presented all the significant differentially expressed biological processes and cellular components (both at gene and protein levels) in the supplementary excel file. Due to the complexity of the data, it is difficult to draw any meaningful conclusion by considering all the significant differentially expressed biological processes and cellular components. Based on our a priori hypothesis (i.e. FCA and SCA differ with regard to hormone production and secretion, and growth potential) we selected several biological processes of relevance to the clinical implications and tumor phenotypes for further detailed analyses. In addition we believe that the figures nicely complement the tables and contribute to a better understanding of the results.
Last, it would be better to provide the molecules associated with the selected BPs in the table. And, it may be helpful to analyze the proteins or genes that act as a hub in the above networks.
Thank you for your suggestion. A detailed description of the molecules associated with the selected BPs is presented in Suppl.File_GO matches_files.
Major points #6
Some contradictions within the paper need to be corrected or discussed. I wonder if the expression of PSCK1, a corticotroph cell marker, showed significant differential expression in the proteomics dataset because it is not included in Table 1 which listed the overlap of DEG and DEP. However, in line 199-200, the authors described as it is increased in proteomic data. Please clarify this. If the expression of proteins, both PSCK1 and PSCK2, are not significantly altered in the proteomics data, please discuss the reason.
Thank you for pointing this out. Please find the figure below to present PCSK1 and PCSK2 gene and protein data in all samples submitted to RNA seq and Proteomic studies.
As presented in the figure, there is a wide biological variability between the different patients potentially leading to a type two error. The only significant results are: 1. PCSK1 is significant different at the protein expression levels between the two groups; and 2. PCSK2 is significant different at the gene expression level between the two groups. However, when measured in a larger cohort of tumors, PCSK1 gene expression was significantly higher in FCA as compared to SCA (as shown in the Figure 1 of the manuscript). As the reviewer correctly noticed, Table 1 includes a list of genes and proteins that overlap (i.e are significantly differentially express both at gene AND protein level). As shown in the figure above none of the PCs has this attributes, thereby not included in Table 1. Thus we have now commented on the biological variability between the different patient samples (and the risk of type 2 error) in the manuscript, as one of the reasons to explain the discrepancy (please see lines 266-269).
The authors mentioned in lines 132-133 that Wnt signaling is more activated in FCA than SCA. However, in the Introduction section, as the authors stated, the more aggressive one is SCA. An explanation for this inconsistency is required.
We do agree with the reviewer on the discrepancy. However, the literature is sparse regarding data on the Wnt signalling (and other developmental signalling pathways) in pituitary tumors. We don’t know if the activation of Wnt signalling in corticotroph tumors will lead to a higher growth potential or may be involved in hormonal production. We do acknowledge the discrepancy and have already discussed it further in the discussion section (lines 338-343): “There was no clear evidence of the presence of consistent differences in the activation of classical developmental signaling pathways (i.e. TGF-beta signaling, WNT signaling, Hippo signaling, Notch signaling) between the groups. However, we identified a few markers to be significantly different. These signalling pathways have often several layers of regulation and often the activation implies phosphorylation of key proteins, so one should be cautious interpreting gene data alone.”
Likewise, the BP “neovascularization” might be a hallmark of SCA because it is aggressive cancer. Why is this BP enriched only in FCA? Conversely, the BP “hormone processing and secretion” seems to be a characteristic of FCA. However, this BP term is enriched in FCA. If this BP term is related to other hormones except for ACTH, please discuss them.
We do agree with the reviewer. Our study is definitely hypothesis generating and further mechanistic studies are needed to understand the results further. Indeed, some proteins involved in “neovascularization” were increased in FCA whereas some proteins involved in hormone processing and secretion were increased in SCA. As discussed throughout the manuscript both subgroups presented different molecules involved in hormone processing and secretion- and mechanistic studies are needed to further elucidate their different role in hormone production.
The proteins found to be increased in the FCA group are involved in the ER protein processing signalling pathway, so they are important for the entire secretory capacity of the cell. We don’t have any evidence for the increase production of “other hormones” by the FCA, but it is clear that the secretory pathway is more active in FCA.
Major points #7
The non-MS methods (e.g. Western blot, ELISA, IHC, RT-PCR, etc.) are needed to support the omics results. The authors suggested that the genes CRABP, LGALS3, GAL as important proteins in the Discussion section. If there are additional experimental results for the expressions of these genes, it will be a concrete basis for the authors' suggestions.
We do agree that other methods are needed to better understand the data. We do not have data on the mentioned proteins but we have tested all the genes involved in ER processing in a large cohort of tumors and found them to be significantly different also when tested in a larger cohort of tumors. The data was presented as an orally poster at ECE 2019.
Minor points #1
I suggest the authors to supplement the contents in lines 122-128 with more details. Simply listing collagen/cell adhesion/extracellular matrix (ECM)-related genes or proteins does not give the readers any information.
Thank you for the suggestion. We added the contents in lines 122-128. Please see the revised manuscript (lines 126-130).
Minor points #2
In lines 134-143, a brief description of each significant protein is written. However, the association between these proteins and PitNET is not described. Please complement detailed information.
Thank you for the suggestion. There is no data in the literature on these genes and their expression in PitNET 134-143.
Minor points #3
There are many paragraphs (or sentences) in this paper that are cut off in halfway of explaining something. For example, the paragraph covered line 187-188 has an interesting introduction like “The ECM proteins cluster covered several collagen genes in addition to LAMA4 and SERPINF1 and SPON2 (Figure 4 A and Suppl.Excel.File_GO_matches FCA_SEQ).”, but the authors instantly stop explain specific things. This significantly interferes with reading. The overall revision of writing is needed.
We thank the Reviewer for the comment and for pinpointing the language mistakes detected. Accordingly, the manuscript has been profoundly revised.

Reviewer 2 Report
The manuscript submitted by Eieland et al. aimed to investigate the differences between FCA and SCA by integrating transcriptome with proteomics data, followed by pathway analyses. The manuscript is nicely written and may be interesting to clinicians managing PitNETs, as well researchers in the field. However, the study has some flaws and limitations that need to be addressed by the Authors in order to improve the quality of their article.
1. The cases subjected to RNA sequencing are few and substantial amount of information is missing from Suppl Table 1 in respect of clinico-pathological and laboratory,radiological parameters, invasiveness/radicality etc. Would the Authors consider including more cases and review pathology-radiology reports, hormonal levels (for FCA) and patient journals? Were there any non-sporadic PitNETs?
2. The results should be presented more clearly and not be discussed and elaborated in the Results section. For example, the text in page 5, lines 129-143 should be moved to the Discussion section. Accordingly, the parts on pathway and network analysis should be shortened. To achieve this,the Authors could elaborate and explain better Figures 3 and 4 in figure legends.
3. Please address better potential limitations of the study in discussion section, including its retrospective nature, small sample size in certain analyses, and e.g. lack of IHC testing of SSR status.
4. The results demonstrate that FCA and SCA show differences, but the clinical and biological relevance of these findings are difficult to interpret. The authors could elaborate more on the implications for therapy/biomarker development of their findings and deliver a more clear message to the reader in the Discussion-Conclusion section. Could the Authors suggest a limited number of biomarkers for further testing based on their analysis?
Author Response
Point-by-point response to the Reviewers’ comments
We sincerely thank the Reviewers for their constructive comments, which we found very helpful towards improving the quality of our manuscript. Accordingly, specific changes have been made in the manuscript, based on these comments, as it is described in detail below in a point-by-point response to the reviewers. Changes within the manuscript are highlight in yellow.
RESPONSES TO Reviewer 2
The manuscript submitted by Eieland et al. aimed to investigate the differences between FCA and SCA by integrating transcriptome with proteomics data, followed by pathway analyses. The manuscript is nicely written and may be interesting to clinicians managing PitNETs, as well researchers in the field. However, the study has some flaws and limitations that need to be addressed by the Authors in order to improve the quality of their article.
- The cases subjected to RNA sequencing are few and substantial amount of information is missing from Suppl Table 1 in respect of clinico-pathological and laboratory, radiological parameters, invasiveness/radicality etc. Would the Authors consider including more cases and review pathology-radiology reports, hormonal levels (for FCA) and patient journals? Were there any non-sporadic PitNETs?
Thank you for the suggestion. We have considered it and have added a supplementary figure presenting the typical MRI characteristics of FCA and SCA. In addition, we have supplemented the Suppl. File. Table S1 with available data on Ki67, ACTH and Cortisol levels for all the included patients.
- The results should be presented more clearly and not be discussed and elaborated in the Results section. For example, the text in page 5, lines 129-143 should be moved to the Discussion section. Accordingly, the parts on pathway and network analysis should be shortened. To achieve this,the Authors could elaborate and explain better Figures 3 and 4 in figure legends.
Thank you for the suggestion. We have now elaborated on the legends to Figure 3 and 4. We do agree with the reviewer that the lines 129-143 may be included in the discussion section. We have cited (in the results section) some of the relevant previous work describing the genes found to be different in our dataset. We believe that the description of the genes is important to be presented in the results section for readability. For the sake of clarity, we choose to let the small description regarding the previous work in the results section. However if the reviewer and/or the editor want to move it to the discussion part of the manuscript we’ll gladly do so.
Please address better potential limitations of the study in discussion section, including its retrospective nature, small sample size in certain analyses, and e.g. lack of IHC testing of SSR status.
Thank you for the suggestion. We have addressed some of the limitations of our study in the end of the discussion section (lines 364-361). Regarding the SSR status: None of the SSRs were differentially expressed at gene and protein levels, so for our data the SSRs status seems not to be a reliable criteria in order to differentiate the functioning vs. the silent corticotroph tumors.
The results demonstrate that FCA and SCA show differences, but the clinical and biological relevance of these findings are difficult to interpret. The authors could elaborate more on the implications for therapy/biomarker development of their findings and deliver a more clear message to the reader in the Discussion-Conclusion section. Could the Authors suggest a limited number of biomarkers for further testing based on their analysis?
Thank you for your comment. We have now added information on the possible biomarkers as revealed in our study (please see the revised manuscript).

Round 2
Reviewer 1 Report
The authors made effort to complement the previous version of manuscript. Through the revision of some parts, the understanding of the paper became much easier. However, most of my major concerns were not addressed. In particular, the network analyses for transcriptome and proteome are still not informative. In addition, the discussion about the findings in the study was still insufficient and descriptive. Finally, the results of validation were not included in the revised manuscript although some results of RT-PCR are provided in the response letter. I think this paper is still showing preliminary results and additional revision process is necessary.
Author Response
Response to Review 1, 2nd round
The authors made effort to complement the previous version of manuscript. Through the revision of some parts, the understanding of the paper became much easier. However, most of my major concerns were not addressed.
In particular, the network analyses for transcriptome and proteome are still not informative.
Thank you for pointing this out again. We have now increased the quality of the figures (Figure 3, 4, 5) as presented in the manuscript.doc (please see the revised manuscript). We have realised that the figures (3, 4, 5) presented in the manuscript.doc file are of poor quality and difficult to read. During the submission process we have provided high resolution .tif files of all the figures together with the supplementary files, but we are not sure if these were available to the reviewer during the reviewing process (we apologise for this, but we assumed that the higher resolution figures were available for the referees).
In addition to update the figures to a higher resolution in the manuscript.doc file, we have now further emphasized that the figures have a user friendly interactive online version were the reader can get a detailed overview of the results.
For example: If the reader wants to evaluate the DEP presented by the FCA in an interactive manner, this can be done by clicking the link: https://version-11-0.string-db.org/cgi/network.pl?networkId=nBtGbCu86iDj. Here, you can choose to visualise, in an interactive manner, under the “analyses” rubric the “Functional enrichments in your network”: Biological Process (GO); Molecular Function (GO); Cellular Component (GO); local STRING network cluster; KEGG Pathways; Reactome Pathways, etc. The main conclusions and the discussion of the manuscript are presented in the main body of the manuscript. But the reader may draw their own conclusions and (for example) look themselves for their gene/protein of interest. The “omnics” data are large and we do provide the reader with our “own conclusions”, but also with the possibility to test their own hypothesis, if wanted.
Figure 3, presents all the DEG in both subgroups of adenomas, the dots are proportional with the log2 fold of change and the relationships between all the genes are presented. We invested considerable efforts in creating this figure (using R programming) and made it available (https://alexeie.github.io/corticotroph-adenomas/) for the ease of the reader. The presentation of the data in this manner is novel in the literature. In all available visualisation programs, the users have to separate up- and down-regulated genes manually, and process them in separate analyses. Thereby, any result is only a partial representation of the underlying biological changes. In addition, the traditional figures do ignore any underlying quantitative information.
We have now updated Figure 3 in the manuscript and described the novelty of the data presentation in this way in the discussion section. We do hope the reviewer will browse the on-line version of the figure and appreciate its usefulness.
In addition, the discussion about the findings in the study was still insufficient and descriptive.
We have now restructured the discussion section. As noted in the discussion, this is a hypothesis generating/descriptive study and mechanistic studies and validation in larger cohorts are needed, in order to test the raised hypothesis. Nevertheless, despite the relative small numbers of patients included in the study, the results were consistent (at least in the RNAseq study where the tumours do group according to their clinical phenotype). Moreover, data with regard to POMC related genes, ER protein processing, and cellular adhesion molecules were present in both omnic studies, thereby reinforcing the results. By this, we raise two original and novel hypothesis, of highly relevance for future treatment perspectives of these orphan patients (1. the role of ER processing proteins in ACTH production/secretion, 2. the different profile of cell adhesion molecules presented by these adenomas).
Finally, the results of validation were not included in the revised manuscript although some results of RT-PCR are provided in the response letter.
We do appreciate this comment. Our understanding is that RNAs seq. technology has become so exact that the RT-PCR validation of the results in the same adenomas should not be necessary. However, confirming data in a larger cohort of adenomas, in order to straighten the clinical significance is definitively important. We have included data on POMC, TBX19 and PCSK1/3 in a larger cohort of patients. The results regarding ER protein processing (included in the previous response letter) as presented as a poster at ECE 2019 are now submitted as a supplementary file and a comment is added in the result section of the manuscript.
I think this paper is still showing preliminary results and additional revision process is necessary.
We thank the reviewer for many constructive comments that lead to a considerable improvement of the quality of our manuscript. We hope that the reviewer finds the new version of the manuscript worth publishing in Cancers.

Round 3
Reviewer 1 Report
The authors said “The figure 3 presents in a novel manner simultaneously the relationships between all the DEGs in both subgroups of adenomas. As shown, the relations between differentially expressed genes are complex and drawing clear conclusions on the clinical significance is difficult.” in lines 162-164. However, if this network analysis only leads to the conclusion that the DEG network is complex and therefore difficult to interpret, its importance is seriously diminished. I understand the efforts the authors have made for this interactive network, but I still have concerns about how it can help the potential readers. Of course, I have experienced this interactive network through the URL provided by the authors. In the network, two groups of genes are just being displayed (even a co-expression module was not indicated) and one who wants to see what gene is in the network should search it in the drop-down list. For example, when I selected MAPK1, many genes interacting with MAPK1 and their fold change were highlighted. However, the MAPK1-related genes themselves, which are dysregulated in FCA and SCA, cannot provide useful information. The difficulty of deriving any biological/clinical meaning from this network is evident, as the authors have mentioned. I want the authors to consider once again why this network is needed in the current paper.
In both Figure 4 and Figure 5, labels of the nodes (genes or proteins) in networks are too small to read. Please increase the font size of the labels to make it easier to read. Furthermore, in the current form, the potential readers may find it difficult to identify what gene/protein module is associated with the GO terms mentioned by the authors. I inquire the authors to overlay all the GO terms in Figure 4C and 4D (or some representative functions mentioned in text) on the network and to do the same thing to the proteome networks. Also, please add representative GO terms from the proteomics data in Figure 5 as tabulated form as the authors have done for Figure 4C and 4D. Last, please specify the statistics from the ORA in the Tables, such as Figure 4C and Figure 4D (and additional Tables in Figure 5).
